# Rapid prediction of crucial hotspot interactions for icosahedral viral capsid self-assembly by energy landscape atlasing validated by mutagenesis

**Ruijin Wu**, **Rahul Prabhu**, **Aysegul Ozkan**, **Meera Sitharam***

Department of Computer and Information Science and Engineering, University of Florida, Gainesville, Florida, United States of America

* sitharam@cise.ufl.edu

**Data Availability Statement:** Supporting information including raw prediction data and code

## Abstract

Icosahedral viruses are under a micrometer in diameter, their infectious genome encapsulated by a shell assembled by a multiscale process, starting from an integer multiple of 60 viral capsid or coat protein (VP) monomers. We predict and validate inter-atomic hotspot interactions between VP monomers that are important for the assembly of 3 types of icosahedral viral capsids: Adeno Associated Virus serotype 2 (AAV2) and Minute Virus of Mice (MVM), both $T = 1$ single stranded DNA viruses, and Bromo Mosaic Virus (BMV), a $T = 3$ single stranded RNA virus. Experimental validation is by in-vitro, site-directed mutagenesis data found in literature. We combine ab-initio predictions at two scales: at the *interface-scale*, we predict the importance (*cruciality*) of an interaction for successful subassembly across each interface between symmetry-related VP monomers; and at the *capsid-scale*, we predict the cruciality of an interface for successful capsid assembly. At the interface-scale, we measure cruciality by changes in the capsid free-energy landscape *partition function* when an interaction is removed. The partition function computation uses *atlases* of interface subassembly landscapes, rapidly generated by a novel geometric method and curated opensource software EASAL (efficient atlasing and search of assembly landscapes). At the capsid-scale, cruciality of an interface for successful assembly of the capsid is based on combinatorial entropy. Our study goes all the way from resource-light, multiscale computational predictions of crucial hotspot inter-atomic interactions to validation using data on site-directed mutagenesis' effect on capsid assembly. By reliably and rapidly narrowing down target interactions, (no more than 1.5 hours per interface on a laptop with Intel Core i5-2500K @ 3.2 Ghz CPU and 8GB of RAM) our predictions can inform and reduce time-consuming in-vitro and in-vivo experiments, or more computationally intensive in-silico analyses.

are available at https://geoplexity.bitbucket.io/virusSuppInfo.html.

**Funding:** The research in this manuscript was supported in part by the following National Science Foundation (NSF, https://www.nsf.gov/) grants awarded to author Meera Sitharam, DMS-0714912, CCF-1117695, DMS-1563234, and DMS-1564480. The funders had no role in study design, data collection and analysis, decision to publish, or preparation of the manuscript.

**Competing interests:** The authors have declared that no competing interests exist.

## Author summary

Viruses, found in all classes of living orgaisms, can be beneficial as well as harmful to their hosts. Understanding their mechanism of assembly is critical to understanding how we can inhibit or enhance their life cycle process. Icosahedral viral capsids, as elucidated by Caspar and Klug, are self-assembled from nearly identical viral capsid or coat-protein (VP) monomers spontaneously and rapidly, with high efficacy and accuracy, a process sometimes facilitated by other biomolecules. Understanding virus assembly requires identifying crucial VP-VP hotspot interactions whose removal would disrupt the process. We combine a novel geometric method for rapidly atlasing free energy landscapes with a symmetry-based combinatorial method to give a two-scale prediction of hotspot interactions. We validate the predictions for 3 types of viruses, using in-vitro, site-directed mutagenesis' disruptive effects on capsid assembly, found in literature, noting that the biophysical assays for AAV2 were carried out by the Mavis Agbandje-Mckenna's lab contemporaneously with the development of our computational model and prediction. Our predictions are reproducible using our curated opensource software EASAL (efficient atlasing and search of assembly landscapes). To the best of our knowledge, prevailing methods for statistical mechanical prediction of hotspot interactions use a single scale, are knowledge-based, are computationally intensive, or have not been validated by in-vitro site directed mutagenesis results.

## Introduction

Viruses can be pathogenic or non-pathogenic, rod-like or icosahedral, enveloped or non-enveloped. Pathogenic viruses are detrimental to their host and significant research is focused on their prevention, by disrupting crucial steps in their life cycle [1]. Virus capsid assembly is a critical step in the generation of infectious virus particles during their replicative life cycle. Understanding assembly processes in the viral life cycle illuminates the pathophysiology of infectious diseases, and allows us to target assembly processes with drugs. Improving assembly of non-pathogenic viruses can be utilized for certain beneficial applications, for example cancer treatment with oncolytic viruses, cell and gene therapy applications, and for vaccine production [2].

Icosahedral viral capsids assembled from almost identical viral capsid or coat protein (VP) monomers were elucidated by Caspar and Klug [3]. The number of VP monomers is some multiple (called the *T* number) of 60. At each inter-monomeric, symmetry-related interface, the assembly is a nanoscale process influenced by inter-atomic interactions, while the entire capsid can be between 10's to 100's of nanometers in diameter, involving 100's of interfaces, making capsid assembly a multiscale process. While several aspects of icosahedral capsid self-assembly have been studied in detail [4, 5], its multiscale aspect still remains poorly understood.

Like most other supramolecular assemblies that occur widely in nature, viral capsid self-assembly is extremely robust, rapid, and spontaneous. Spontaneity makes it difficult to control in vitro, rapidity makes it difficult to get snapshots of the process, and robustness makes it difficult to isolate crucial combinations of assembly-driving inter-atomic interactions (see Fig 1).

Assembly involves two types of interactions: (i) the viral coat protein (VP) interactions and (ii) the VP-genome interactions. Although the genome or other biomolecules could influence the VP-VP interaction during the first step of capsid assembly, understanding VP-VP interactions and the VP intermediates generated, in system not requiring additional input, can

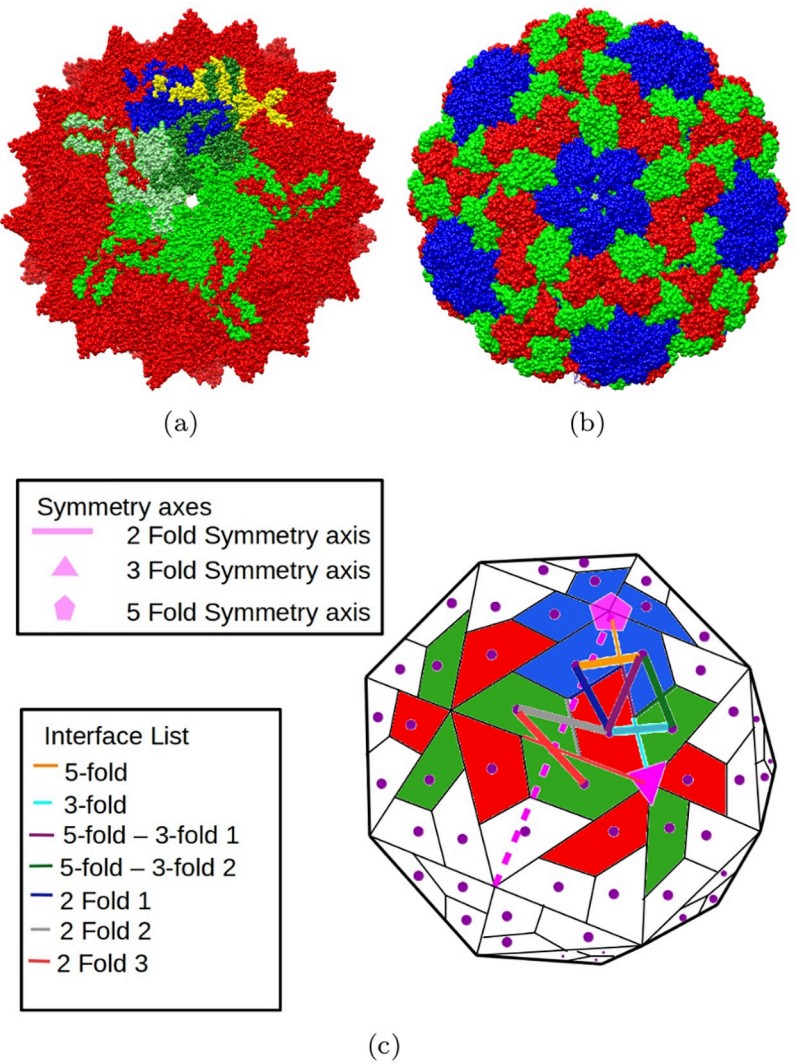

**Fig 1. Structures of a *T* = 1 and *T* = 3 viruses, and a cartoon showing the types of VP monomers and interfaces in the former.** (a) X-ray structure of AAV2 (a *T* = 1 virus). All VP monomers are identical, and the VP monomers colored using the non-dominant colors are used only to highlight the 3 types of interfaces. VP monomers at the 5-fold interface are colored shades of green, light green and blue form a 2-fold interface assembly, and dark green, blue and yellow pairwise form 3-fold interface assemblies. (b) X-ray structure of BMV (a *T* = 3 virus) showing 3 types of VP monomers (green, blue, and red). (c) A cartoon of a *T* = 3 virus showing the 3 types of VP monomers (green, blue, and red), 7 types of interfaces, and 3 symmetries (shown in pink). See the introduction.

inform the utilization of viruses for beneficial applications or the generation of assembly inhibitors that disrupt the formation of pathogenic viruses. Consequently, the self-assembly of several types of icosahedral, non-enveloped viral capsids from identical VP monomers is to date an area of major interest.

A key component in understanding the virus assembly process is identifying those *crucial hotspot interactions* whose removal disrupts assembly. Experimental approaches used to measure the forces involved in determining or orchestrating the VP-VP interaction of the assembled virus include cryo-electron microscopy and image reconstruction, X-ray crystallography, and a variety of quantitative interaction proteomic methods [6] which provide high resolution information about the purified capsid in the crystalline and aqueous states respectively [7],

complete list found on VIPERdb (https://urldefense.proofpoint.com/v2/url?u=http-3A__
viperdb.scripps.edu&d=DwIGaQ&c=sJ6xIWYx-zLMB3EPkvcnVg&r=kGsKDXNcJq1WRG
evTkYaLhTe8S0Zrq5pLMzpMb45Vy0&m=RJ2zjPeU5q0XB4tHkgFcey-Z0oiqNxBsos
EhffocKHs&s=jGUYBEB7WJuz-3UVMaAiQ9g8N_TjuqBf9qIbDULVhVY&e=). These high-
resolution structures can be used to select residues that are conserved within the virus genus
or family and located within symmetry-related interfaces of the icosahedron. *Site-directed
mutagenesis* of the VP followed by gel filtration, light scattering, or sedimentation coefficient
to measure the size of the VP oligomer and to determine the effect of the mutagenesis on
capsid assembly [8]. Other methods used to verify capsid assembly include native capsid
immunoblot or enzyme-linked immunosorbent assay (ELISA), and sometimes cryo-electron
microscopy and other techniques for measuring sizes and concentrations of subassemblies.
These methods of capsid assembly prediction and verification are time consuming and expen-
sive. Additionally, the predictions may not yield mutants that are critical to the process. Thus,
there is a need for rapid and reliable mathematical and computational tools for modeling
supramolecular assembly that can inform further experimentation, including resource inten-
sive in-silico experimentation using computational alanine scanning (CAS) or fine-grained
molecular dynamics (MD) that have to be scaled up from the protein-protein interface level to
the capsid level consisting of at least 150 interfaces.

*Contribution.* The strong influence of entropy contributes to the poorly understood statisti-
cal mechanics of the capsid assembly process, whose free-energy landscape arises from the sys-
tem of inter-atomic interactions at interfaces between the nearly identical VP monomers.
Icosahedral symmetry restricts the interface types to a small set.

To predict the importance (*cruciality*) of a specific inter-atomic interaction at an interface
for successful capsid assembly, we analyze the viral capsid assembly landscape at two scales,
the *interface-scale* and the *capsid-scale*. At the interface-scale, we measure the cruciality of an
interaction for successful subassembly across an interface type by approximating the changes
in the partition function when the interaction is removed (discussed in the background section
on configurational entropy). We use two *measures* of change in the partition function. The
first uses the partition function for all the minimal energy regions, representing all the stable
subassembly configurations. The second uses the normalized partition function for the poten-
tial energy basin corresponding to the specific subassembly configuration occurring in the suc-
cessfully assembled capsid. This estimates the probability that a stable configuration is in fact
the successful subassembly configuration. We use the ratio of each of these quantities with and
without an interaction—averaged over a principled selection of small subassemblies across an
interface type—to measure cruciality of that interaction for that interface type (discussed in
the section on output of cruciality prediction).

Both measures of change in partition function are rapidly approximated as a *bar-code* that
abbreviates the *atlas* of the interface assembly landscape. The atlas is generated—with minimal
sampling—by the geometric method and curated opensource software EASAL (efficient atlas-
ing and search of assembly landscapes [9–11]). The input to EASAL consists of (a) the VP
monomer geometry—atom coordinates; and for each interface type, (b) pair-potentials for a
candidate set of assembly-driving interactions along with Van der Waals sterics, and (c) small
subassembly structures extracted from known capsid structures. An *atlas* is a partition of the
assembly landscape into contiguous region of nearly equipotential energy called *active con-
straint regions* or *macrostates* (discussed in the section on entropy computation), organized as
a refinable, queryable roadmap, that can further be abbreviated as a *bar-code*. The constraints
are the pair-potentials as in (b) above. The active constraint graphs are analyzed using combi-
natorial graph rigidity, whereby the effective dimension of a macrostate becomes a proxy for

its energy level. The methodology gives fast, light-weight algorithms (100 to 1000 times faster than prevailing methods [11–13]) with rigorously proven accuracy-efficiency tradeoffs.

We additionally give two *types* of predictions, each validated by in-vitro, site-directed mutagenesis results found in literature [14–20]. We note that the biophysical assays for AAV2 were carried out by the Mavis Agbandje-Mckenna's lab [20] contemporaneously with the development of our computational model and prediction. Our first direct, ab-initio prediction generalizes an interface-scale prediction to the capsid-scale by assuming equal importance of each type of interface for capsid assembly and without explicitly accounting for kinetics. *Despite this assumption, and despite the prediction being completely blind to the in-vitro site-directed mutagenesis data* used for validation, this direct, interface-scale prediction correlated well with site-directed mutagenesis data towards capsid assembly disruption in 3 viruses (see the figures in the section on validating the first two-scale prediction).

Our second prediction additionally incorporates a capsid-scale prediction of the cruciality of an interface for capsid assembly. The dimension of the capsid assembly landscape—involving several VP monomers (60 for $T = 1$ and 180 for $T = 3$)—makes direct computations intractable. Hence, we treat the capsid as being recursively assembled from stable subassemblies at interfaces [21, 22], where subassemblies are intermediate oligomeric structures of the capsid, assembled at interfaces (subassemblies are formally defined in the background section on combinatorial entropy). The likelihood of such an *assembly tree*, given successful capsid assembly, is a measure of combinatorial entropy (discussed in the background section on combinatorial entropy). This depends both on the stability and formation rates of the intermediate subassemblies, and the number of equivalent assembly trees under icosahedral symmetry [23–25]. The cruciality of an interface for successful capsid assembly is then determined by all the assembly trees that involve that interface. The relative weights of the cruciality measures described above—the bar-code measuring change of partition function at the interface-scale, and the combinatorial entropy at the capsid-scale—are then determined by statistical learning (discussed in the section on interface cruciality). The learning algorithm uses—for training—a small fraction of the mutagenesis and biophysical assay data towards assembly disruption, to learn the parameters of the statistical model. The remainder of the mutagenesis data is used to validate the cruciality of residues for capsid assembly of 3 types of viral capsids, Adeno Associated Virus serotype 2 (AAV2), which is non-pathogenic, and Minute Virus of Mice (MVM), which is pathogenic to mice, both $T = 1$ single stranded DNA viruses, and Bromo Mosaic Virus (BMV), a $T = 3$ single stranded RNA virus, pathogenic to both monocotyledon and dicotyledon plants.

Our predictions are reproducible using our curated opensource software EASAL (efficient atlasing and search of assembly landscapes https://urldefense.proofpoint.com/v2/url?u=http-3A__bitbucket.org_geoplexity_easal&d=DwIGaQ&c=sJ6xIWYx-zLMB3EPkvcnVg&r=kGsKDXNcJq1WRGevTkYaLhTe8S0Zrq5pLMzpMb45Vy0&m=RJ2zjPeU5q0XB4tHkgFcey-Z0oiqNxBsosEhffocKHs&s=UvqE7o05ehbIXBe1Sgt920eHlxMg3vQCOuBWk0QU0l4&e=, see also video https://cise.ufl.edu/~sitharam/EASALvideo.mpeg, and user guide https://urldefense.proofpoint.com/v2/url?u=https-3A__bitbucket.org_geoplexity_easal_src_master_CompleteUserGuide.pdf&d=DwIGaQ&c=sJ6xIWYx-zLMB3EPkvcnVg&r=kGsKDXNcJq1WRGevTkYaLhTe8S0Zrq5pLMzpMb45Vy0&m=RJ2zjPeU5q0XB4tHkgFcey-Z0oiqNxBsosEhffocKHs&s=AoV5PriolpjfwaF8CxB19gyo8W-Lzbom7Ci4_jTl1VQ&e=). Fig 2 summarizes our overall approach.

Overall, the emphasis of this paper is not the comparison of our interface-scale or capsid-scale predictions with prevailing methods for each individual scale. Rather, our emphasis is on the novel conceptual underpinnings of each of our single-scale predictions, and the validation, using in-vitro site directed mutagenesis data, of our synthesized two-scale predictions. As

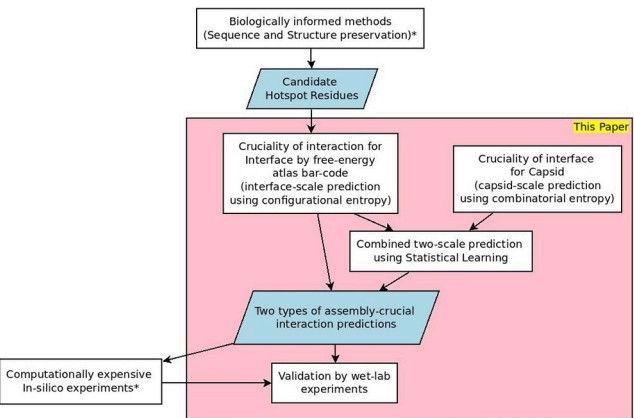

**Fig 2. Flow chart of the methodology in this paper and connections to existing methods.** See the introduction and related work sections.

Fig 2 shows, different aspects of our method can be mixed and matched with prevailing methods to leverage complementary strengths (e.g., for interface-scale hotspot prediction, or extensive in-silico validation).

We are unaware of any previous study that spans the range from multiscale statistical mechanical predictions of crucial hotspot inter-atomic interactions to validation using site-directed mutagenesis results. Previous studies use coarse-grained single-scale, or knowledge-based predictions or use resource-intensive in-silico validation via Computational Alanine Scanning (CAS) or via fine-grained Molecular Dynamics (MD) (see detailed discussion in in the related work section). By reliably and rapidly (taking no more than 1.5 hours per interface on a laptop with Intel Core i5-2500K @ 3.2 Ghz CPU and 8GB of RAM) narrowing down target interactions, our predictions can inform and reduce the time spent on time-consuming in-vitro and in-vivo experiments, as well as more computationally intensive in-silico analyses.

## Related work

At the interface-scale there are several types of methods for predicting crucial hotspot protein-protein interactions (PPI). All of these methods (like ours) use as input a shortlist of candidate hotspot interactions selected using evolutionary sequence or structure preservation. Many of the methods are based on computational alanine scanning (CAS) surveyed recently in [26]. CAS in turn uses Monte Carlo (MC) or Molecular Dynamics (MD) simulations [27–29] to compute binding affinity, free energy or entropic influences of the hotspots. Other methods additionally use a combination of shape specificity, solvent accessibility and prior knowledge-base of PPI via different types of statistical inference or machine learning, see e.g., [30–42]. While exhaustive CAS methods are sometimes validated by site-directed mutagenesis data, most of the hotspot predictions are validated by in-silico CAS experiments, e.g., the SKEMPI database [43]. For example, in a recent paper [44], interface-scale hotspot predictions, combined with specific sequence and structure conservation, have been directly extrapolated to viral capsid scale hotspot predictions. Validation, however, was through computationally intensive all-atom MD sampling. In Fig 2, this approach involves the boxes marked by *.

Free-energy landscapes of protein-protein interface assembly are driven by weak inter-atomic forces and non-covalent bonds and are strongly influenced by the configurational entropy. However, full-blown computation of configurational entropy is a notoriously difficult problem. All prevalent methods for configurational entropy computation rely on computing

the volumes of assembly landscape regions, typically by MC or MD sampling [45–56], which is prohibitive due to the high dimension of the assembly landscape. High geometric or topological complexity of the assembly landscape (disconnectedness, channels of varying effective dimension etc.), means that sampling techniques like MC or MD can only claim stochasticity and uniform sampling in the limit, i.e., when they run for sufficiently long or start from sufficiently many initial configurations [57–62]. Some works such as [51, 63, 64] infer the topology of the configuration space starting from MC and MD trajectories and use topology to guide dimension reduction. On the other hand, methods based on principal component analyses of the co-variance matrices from a trajectory of samples in internal coordinates generally overestimate the volumes of assembly landscape regions. For these reasons, formal accuracy-efficiency tradeoffs are not provided. In contrast, EASAL uses the novel geometric idea of convexifying *Cayley* parameters to represent macrostates, and avoids gradient descent used by the above-mentioned methods, thereby significantly reducing discarded samples and increasing efficiency. Moreover, the EASAL method is able to approximate the configurational entropy of small assembly systems using an atlas bar-code, without relying heavily on sampling, thereby ameliorating the curse of dimension, as shown using rigorous complexity analysis and computational experiments [11]; furthermore formal accuracy-efficiency tradeoff guarantees are provided.

Ab initio methods such as [65], based on geometric algebras are used to give bounds or approximate configurational entropy without relying on Monte Carlo or Molecular Dynamics sampling. However, it is not clear how to extend them beyond restricted assembly systems such as a chain or loop of rigid molecular components, each component consisting of at most 3 atoms, non-covalently bound to their neighboring components at exactly 2 sites. EASAL on the other hand is applicable more generally to assemblies with larger inputs.

While for small assemblies it is possible to atlas the assembly landscape and compute the entropy directly, for larger assemblies, such as virus capsid systems (consisting 60 VP monomers for $T = 1$ and 180 VP monomers for $T = 3$ viruses), the assembly landscape is too big to be atlased directly, although all-atom MD simulations of viral capsid life-cycle processes (docking etc.) post-assembly in the literature, e.g., [66, 67]. Therefore, to tractably deal with the high dimension of their assembly landscape, larger assemblies are typically treated as being recursively assembled as an interface assembly system, from a small number of stable intermediate subassemblies [21].

Several statistical mechanical approaches, as surveyed in [68, 69], could be said to combine configurational entropy and combinatorial entropy into a single scale to analyze kinetics [1, 70–77]. The assembly model [78] based on the local-rules theory [79–82] computes the combinatorial entropy considering both the number of different assembly pathways and the kinetics at each assembly stage. However, such single-scale models rely crucially on the simplified representation of the VP monomers and their geometric interactions, and feature kinetics, rates and concentrations of subassemblies prominently in their analyses. The assembly model in [5] analyzes the efficiency and kinetics of the capsid assembly process. However, they use a simplified coarse-grained model, which assumes that identical capsomeres (pentamers and hexamers) assemble together to form the entire capsid. Similarly, the assembly models in [4, 83, 84] use truncated capsomeres as sub-units of assembly, which when assembled give a perfect icosahedron.

While our method does not *separately* model kinetics, it combines interface-scale and capsid-scale analyses of free-energy, configurational and combinatorial entropy, which could affect kinetics. Furthermore, our method does not rely on a single, capsid-scale analysis with simplified representations of VP monomers, but uses a multiscale analysis.

Several computational studies have been conducted on various aspects of the virus life cycle. The paper [85] uses rigidity analysis on the fully assembled capsids of icosahedral proteins to identify functional units of the capsid. Assembly pathways have been used to study the self-assembly of polyhedral systems from identical sub-units; for example, the paper [86] studies the role of assembly pathways and the degrees of freedom of intermediate subassemblies in the self-assembly of polyhedra with known isomers. The goal is to manipulate the degrees of freedom of the intermediate sub-assemblies to increase the concentration of one isomer over others. In contrast, we use graph and symmetry analysis of assembly pathways of viral capsids with the goal of identifying interfaces that are crucial to assembly. We further use graph rigidity to analyze and synthesize the two scales, namely configurational and combinatorial entropy, of capsid assembly.

**Organization.** The paper is organized as follows. The materials and methods section describes the predictions of cruciality of inter-monomeric interactions for interface assembly and for capsid assembly as a whole. The results section provides the results validating the cruciality prediction of interactions to the capsid, in three viruses, AAV2, MVM, and BMV.

## Materials and methods

In the background section on configurational entropy, we provide some background on the configurational entropy of virus capsid assembly. The section on entropy computation describes key features of the EASAL methodology (see software https://urldefense.proofpoint. com/v2/url?u=http-3A__bitbucket.org_geoplexity_easal&d=DwIGaQ&c=sJ6xIWYx-zLMB3EPkvcnVg&r=kGsKDXNcJq1WRGevTkYaLhTe8S0Zrq5pLMzpMb45Vy0&m=RJ2zjPeU5q0XB4tHkgFcey-Z0oiqNxBsosEhffocKHs&s=UvqE7o05ehbIXBe1Sgt920eHlxMg3vQCOuBWk0QU0l4&e=, video https://cise.ufl.edu/%5C~sitharam/EASALvideo.mpeg, and user guide https://urldefense.proofpoint.com/v2/url?u=https-3A__bitbucket.org_geoplexity_easal_src_master_CompleteUserGuide.pdf&d=DwIGaQ&c=sJ6xIWYx-zLMB3EPkvcnVg&r=kGsKDXNcJq1WRGevTkYaLhTe8S0Zrq5pLMzpMb45Vy0&m=RJ2zjPeU5q0XB4tHkgFcey-Z0oiqNxBsosEhffocKHs&s=AoV5Priolpjfwa F8CxB19gyo8W-Lzbom7Ci4_jTl1VQ&e=) and entropy computation with it. In the background section on combinatorial entropy we discuss the combinatorial entropy in viruses.

In the section on interaction cruciality, we describe the computation of the cruciality of inter-atomic interactions across VP monomers to interface subassembly and thereby to capsid assembly. In the section on interface cruciality, we describe a second scale of cruciality of interfaces to capsid assembly. In the section on two-scale prediction, we describe statistical models to combine the interface-scale configurational entropy and the capsid-scale combinatorial entropy to predict the cruciality of an interaction at the capsid level.

### Background: Configurational entropy in virus assembly

The efficacy of viral capsid assembly is largely due to the structure of its equilibrium free energy landscape. Specifically, the depth and volume of the potential energy basins, including the basin containing the successfully assembled capsid configuration. The free energy at a basin depends on the average potential energy and the configurational entropy of the basin. Of these, the computation of the configurational entropy dominates the computation of free energy.

Let $E(x)$ be the potential energy function, defined over the assembly landscape, for an assembly configuration $x$ (the function $E$ is described in detail in the next section). The

partition function $Q$ is a integral over the energy basin $\beta$, given by

$$Q = \int_\beta e^{\frac{-E(x)}{k_B\,T}}\,\mathrm{d}x$$

where $x \in \beta$ is a configuration in the basin, $k_B$ is the Boltzmann's constant and $T$ is the absolute temperature. The configurational entropy $S$ of the basin is

$$S = k_B \ln Q + \frac{\langle E \rangle}{T}$$

where $\langle E \rangle$ is the the average energy over the basin.

The free energy $F$ of a system with a single energy basin $\beta$ is given by:

$$F = \langle E \rangle - TS$$

Hence, over a region $C$ of constant energy $E_C$, for example an active constraint region as defined in EASAL, the entropy is merely a function of the volume $V_C$ of the region.

$$S_C = k_B \ln V_C \tag{1}$$

where $V_C = \int_C \mathrm{d}x$.

In a landscape with multiple potential energy basins $\beta_i$, each of which has a constant energy $E_i$, the partition function of each energy basin $Q_i$ can be expressed as a weighted sum of the volumes $V_i$ of the different basins.

$$Q_i = \int_{\beta_i} \mathrm{d}x \cdot e^{\frac{-E_i}{k_B T}} = V_i \cdot e^{\frac{-E_i}{k_B T}} \tag{2}$$

The normalized partition function $p_i$ is the probability of finding the system in the energy basin $\beta_i$:

$$p_i = \frac{Q_i}{\sum_i Q_i} \tag{3}$$

In the next section we show how to approximate the computation of the partition function by generating an atlas of the capsid assembly landscape using EASAL and extracting a relevant bar-code.

## Atlasing and entropy computation using EASAL

An *interface assembly system* (see Fig 1) consists of (a) the VP monomer geometry—atom coordinates; and for each interface type, (b) short-range Lennard-Jones potentials for a candidate set of interactions, i.e., atom pairs (one from each VP monomer) along with Van der Waals sterics, and (c) small subassembly structures extracted from successfully assembled capsid.

The potential energy $E(x)$ for an *interface assembly configuration* $x$ has one Lennard-Jones term for each atom pair (one from each VP monomer). In EASAL, the short-range Lennard-Jones pair potentials are *geometrized* by discretizing into three intervals: large distances at which Lennard-Jones potentials are no longer relevant, contributing $E_h$ to the potential energy of the configuration; short distances prohibited by Van der Waals forces; and interval between the two known as the Lennard-Jones *well*, contributing $E_l$ to the potential energy of the configuration. We say that a pair of atoms has an *active constraint* if the distance between their centers is within the discretized Lennard-Jones well.

For a landscape with $N$ Lennard-Jones terms, potential energy of a configuration with $N_a$ active constraints is given by:

$$E = N_a E_l + (N - N_a)E_h = NE_h - N_a(E_h - E_l) \tag{4}$$

In the expression for partition function in Eq 2, each configuration contributes a weight

$$e^{\frac{-E}{kT}} = e^{\frac{-NE_h + N_a(E_h - E_l)}{kT}} = C \cdot \left( e^{\frac{E_h - E_l}{kT}} \right)^{N_a}$$

where $C = e^{\frac{-NE_h}{kT}}$ is a constant of the landscape and is canceled out when calculating the normalized partition function, and the weight

$$w(N_a) = \left( e^{\frac{E_h - E_l}{kT}} \right)^{N_a} \tag{5}$$

With this geometrization of energy, the potential energy basin is completely determined by its partition into *active constraint regions*, i.e., regions of the assembly landscape whose configurations have a particular set of active constraints and hence, nearly constant potential energy. This gives a queryable *roadmap* of the basin, where each region is uniquely labeled by an *active constraint graph*, whose edges are the active constraints and whose vertices are the participating atoms (see Fig 3). Using combinatorial rigidity [87], each active constraint generically reduces the effective dimension of the region by one. The bottom of each basin is a 0-dimensional region $R$, with active constraint graph $G$, containing the minimum energy configurations. The higher energy regions leading to $R$ are exactly those that have active constraint graphs that are subgraphs of $G$.

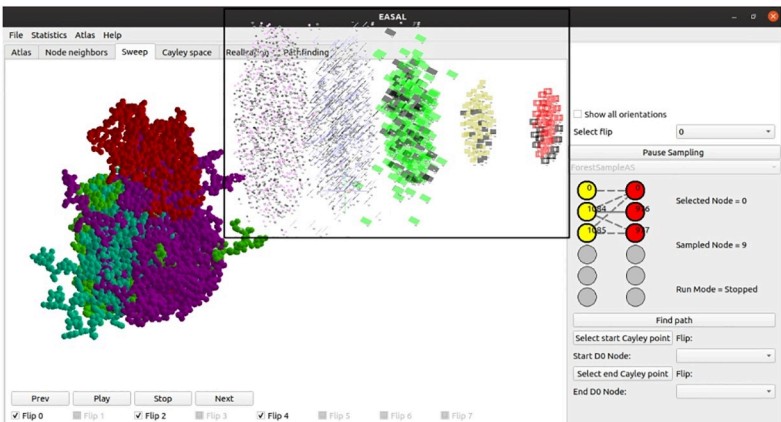

**Fig 3. Screenshot of the EASAL software showing all configurations in an active constraint region in the atlas of the interface assembly system of the two VP monomers shown on top right.** The region's active constraint graph is shown at bottom right, with red and yellow representing atoms in different VP monomers, the single bold edge representing a single active constraint or interaction $c$, and the dashed lines representing the 5 Cayley parameters that are used to convexify this effectively 5-dimensional region. On the main screen, the red VP monomer is held fixed and all of the second VP monomer's relative positions (satisfying the one active constraint $c$) are shown. The 3 different colors (cyan, green and purple) of the second VP monomer sweeps represent distinct orientations within the same active constraint region. (inset) Atlas with changes when an interaction is disabled. Active constraint regions (nodes of the atlas) of different dimensions are shown in different colors, with red nodes representing regions with 2 active constraints, or 4 effective dimensions, and each of the successive strata (from right to left) showing regions of one more active constraint, or one lower energy level or effective dimension. The left most are the 0-dimensional or lowest energy regions, each of which is the bottom of a potential energy basin, with all its ancestor regions participating in the basin. The black nodes are the active constraint regions that disappear from the atlas due to the removal of a candidate inter-atomic interaction. See the section on entropy computation and the section on interaction cruciality.

One of EASAL's key features is the generation of roadmaps for all basins, called an *atlas*, without relying heavily on sampling. This is achieved by using a recursive method that searches the interior of higher energy regions for boundary regions with exactly one new active constraint. Searching for such boundary regions (which are effectively of one fewer dimension) has a higher chance of success than directly looking for the lowest energy regions, which are the lowest dimensional active constraint regions.

Staying within active constraint regions is achieved by a second key feature of EASAL: *convexifying* active constraint regions using customized, distance-based or *Cayley* parametrization, avoids gradient-descent to enforce active constraints, and results in high efficiency search with minimal sampling and reduced repeated or discarded samples. In addition, it is straightforward to compute the inverse map from the Cayley parameter values to their corresponding finitely many Cartesian configurations. Altogether, EASAL obtains comparable coverage with 100 to 1000 times fewer samples than prevailing methods [11–13]. Cayley convexification leverages geometric features that are unique to assembly (as opposed to protein folding). Together, the active constraint regions, their effective dimensions, and their volume approximations obtained through Cayley parameterization, provide an abbreviated atlas *bar-code* for the basin structure of the assembly landscape.

## Background: Combinatorial entropy in virus assembly

Combinatorial entropy of capsid assembly captures the number of possible ways in which a successful assembly configuration can be recursively decomposed into subassemblies down to the rigid motifs in the VP monomers [21, 23–25]. In reverse, larger assemblies are treated as being recursively assembled as interface assembly systems. Since the VP monomers that are far away from the interface tend to have little impact on the assembly, we can simplify the participants of each interface assembly system to VP monomers or dimers near the interface.

As shown in Fig 4, there is typically more than one way of treating a subassembly as an interface assembly. When there are multiple interfaces to choose from, we consider the free energy and reaction rates of each of the options and pick the best interface for the subassembly.

With this setup, we define a labeled binary tree, called an *assembly tree*, to describe how a series of subassemblies leads to a full capsid assembly. In an assembly tree, the root node is a successfully assembled viral capsid, and the leaves are VP monomers. Every internal node of the tree is a subassembly, labeled by its best interface (as defined earlier). Fig 4, shows an assembly tree for a $T = 3$ viral capsid. Given the free energy and reaction rate of each subassembly and the structure of the assembly tree, we can define its likelihood under the assumption of successful assembly.

An *assembly pathway* is a collection of assembly trees that satisfy some prediction-related criteria [23–25]. For example, all assembly trees that are in one equivalence class under icosahedral symmetries can be grouped as a single assembly pathway. As another example, an assembly pathway can be defined as the collection of such symmetry classes that do not use specific types of interfaces. The papers [23–25] enumerate assembly pathways and compute their likelihood for such criteria.

## Interaction cruciality at interface-scale

We use the atlas generated by EASAL to compute two quantities for each interface assembly landscape: (a) the partition function for minimal energy regions (basin bottoms), and (b) the normalized partition function for the potential energy basin corresponding to the known

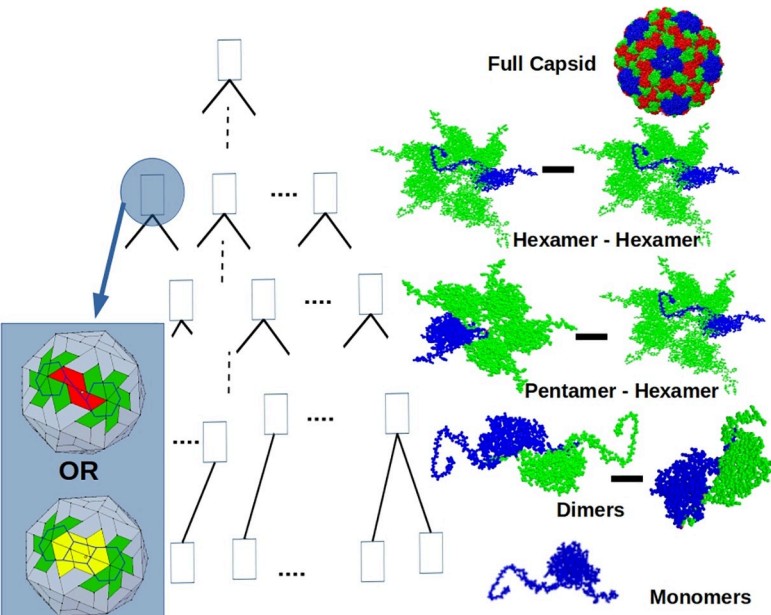

**Fig 4. An assembly tree of a *T* = 3 viral capsid.** The root node represents a successfully assembled viral capsid. Each internal node represents an interface assembly system that contains a stable subassembly configuration that is part of the known, successfully assembled capsid configuration. Children of a node are the participating multimers for the node's interface assembly system. The leaf nodes represent the VP monomers. To the right of the nodes are their candidate stable subassembly configurations taken from the *T* = 3 BMV X-ray capsid structure. At internal nodes, a choice is made between multiple candidate interface assembly systems. On the left we highlight an internal node with 2 available choices for hexamer-hexamer interfaces, of which one is chosen: the inset shows the choices—a single VP dimer interface highlighted in red; and two VP dimer interfaces, highlighted in yellow. See the background section on combinatorial entropy.

(successful) interface subassembly configuration called the *true realization*. These two parameters serve as an atlas bar-code to determine the cruciality of interactions at the interface-scale.

As mentioned earlier, the bottom of each basin is a 0-dimensional region *R*, with active constraint graph *G*, containing the minimum energy configurations. The higher energy regions leading to *R* are exactly those that have active constraint graphs that are subgraphs of *G*. Fig 5 illustrates, using EASAL screenshots, the basin structure of two VP monomers assembling across a hexamer interface in BMV.

Two assembly configurations are considered distinct if and only if their *similarity* distance (the 2-norm distance between their point coordinate vectors) is at least $\varepsilon$. The number of distinct Cartesian configurations in a region then becomes an approximate measure of the size or volume of the region (configurational entropy associated with that region).

For any interface assembly system *s*, since the energy of all 0-dimensional configurations is the same, we approximate the sum of the $Q_i$'s in Eq (2), with the number of distinct configurations in the union—denoted by $R_0^s$—of all the 0-dimensional active constraint regions. Formally, the *partition function for all minimal energy regions* of the atlas of a given interface assembly system *s* is denote it by

$$v_{minima}^s := |R_0^s| \tag{6}$$

The approximation to the normalized partition function of Eq (5) is the ratio of the number of distinct 0-dimensional configurations in the basin of the true configuration (we call this set $R_{true}^s$) to $v_{minima}^s$. This approximates the probability that the assembly process ends in the true

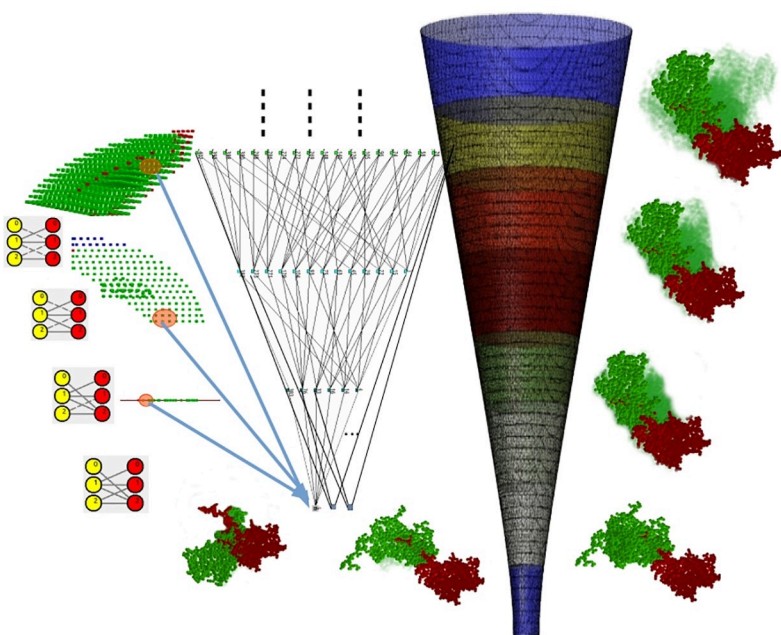

**Fig 5. Prediction using *cruciality bar-codes* described in the section on interaction cruciality for two VP monomers assembling across a hexamer interface in BMV.** Each node in the atlas roadmap in the middle represents an active constraint region (macrostate) in EASAL. Example active constraint graphs are shown at far left: the yellow and red circles represent atoms participating in active constraints (interactions) in the two VP monomers. At each successive level, the number of active constraints increases by 1 and the energy level and effective dimension decrease by 1. The atlas nodes in the bottom-most row represent the 0-dimensional, lowest energy, stable assembly configurations; example configurations shown below them. Their total number (for a given interface $s$, on removal of a given interaction or constraint $r$) gives $v_{minima}^{r,s}$ in the computation of the cruciality bar-code. Each such configuration together with nearby higher-energy configurations in all of their ancestor nodes constitute one potential energy basin. Their sum, across all basins, weighted by energy level gives the denominator of $v_{capsid}^{r,s}$. The rightmost of the stable assembly configurations at the bottom corresponds to the *true realization*. Above it, the 3 solid configurations and the transparent sweeps around them show the closest configurations to the true realization in successively higher energy regions in its basin (one region each for 3 energy levels shown). To the far left, these sweeps are shown as orange highlights in the corresponding Cayley parameterized regions. The colorful basin plot shows the total weighted configurations in the true basin, stratified by dimension or energy level. Their sum is the numerator of $v_{capsid}^{r,s}$.

configuration. To improve this approximation we weight each configuration $x$ inversely to its proximity to a higher energy region $A$ by the weight $w(N_a(x))$ of Eq (5), where $N_a(x)$ is now the number of active constraints in the configurations in $A$.

Thus the normalized partition function for the potential energy basin corresponding to a successful interface assembly configuration is computed using Eq (3) as follows:

$$v_{capsid}^s := \frac{\sum_{\mathbf{x} \in R_{true}^s} w(N_a(\mathbf{x}))}{\sum_{\mathbf{x} \in R_0^s} w(N_a(\mathbf{x}))} \tag{7}$$

Finally, these quantities are used to define our measure of *cruciality of a given input inter-atomic interaction $r$* for a given interface assembly system $s$ to result in a given true configuration. First we define $v_{minima}^{r,s}$ and $v_{capsid}^{r,s}$ as the same quantities in Eqs (6) and (7), respectively, obtained by restricting to a portion of the atlas, i.e., those regions where $r$ is not an edge in the

active constraint graph (see Figs 3 and 5). Now, the *cruciality bar-code* is defined as:

$$\left( \mu_{minima}^{r,s} := \frac{v_{minima}^{r,s}}{v_{minima}^{s}}, \mu_{capsid}^{r,s} := \frac{v_{capsid}^{r,s}}{v_{capsid}^{s}} \right) \tag{8}$$

**Accounting for multimers assembling at an interface.** In a capsid assembly tree, the sub-assembly at an interface could involve either a VP monomer pair or a multimer pair. Although the pair potentials at the interface are specified between the VP monomers closest to the interface, each VP monomer could be part of a multimer whose atoms influence the interface assembly landscape through Van der Waals sterics. We have found that for larger multimers the steric contribution from the VP monomers far from the interface is negligible and that it is sufficient to consider those interface assembly systems involving certain VP monomer-dimer pairs selected as follows.

The *dual graph* of a virus capsid is obtained from the icosahedrally symmetric *capsid polyhedron*, with one face per VP monomer, where interfaces are represented by adjacent faces (see Fig 6). There is one vertex of the dual graph corresponding to each face of the capsid polyhedron and an edge between two vertices if the corresponding faces share an interface.

For the interface represented by the edge *ab* in the dual graph, we consider each triangle *abc*, and generate 3 atlases with the following assembly systems *s*: (i) with VP monomers *a* and *b*, (ii) VP dimer *ac* and VP monomer *b*, (iii) with VP dimer *bc* and VP monomer *a*. For the three $T = 1$ interface types, this gives 9 assembly systems, and for the seven $T = 3$ interface types, this gives 31 assembly systems.

Now, $\mu_{minima}^{r,s}$ and $\mu_{capsid}^{r,s}$ are computed using the atlases for the 3 assembly systems *s* for the same interface *ab*, and then averaged to get cumulative values. These are denoted $\mu_{minima}^{r}$ and $\mu_{capsid}^{r}$ and are used to measure the cruciality of the interaction *r* to the interface *ab*.

## Interface cruciality at capsid-scale

As mentioned in background section on combinatorial entropy, given the free energy and reaction rate of each subassembly of all the nodes and the structure of the assembly tree, we can define its likelihood under the assumption of successful assembly and we can group assembly trees into assembly pathways based on some prediction-related criteria [23–25]. To simplify our model, we abbreviate the notion of the assembly pathway, to a *connectivity pathway*, which only requires a test of connectivity for the internal nodes of the assembly tree.

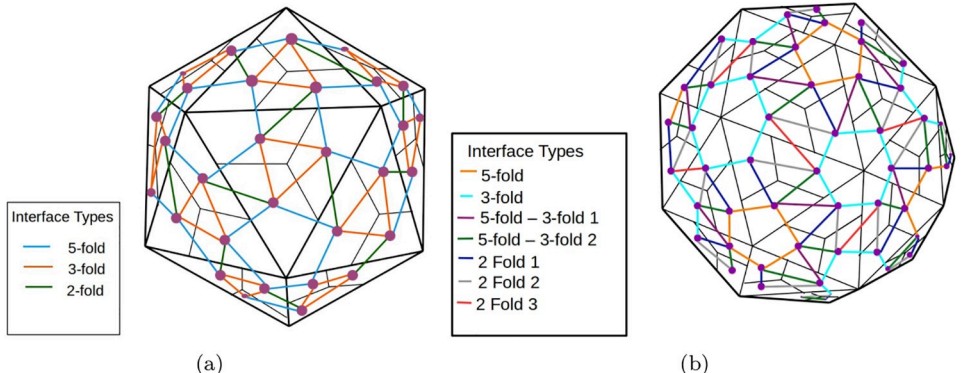

**Fig 6. The dual graphs of $T = 1$ and $T = 3$ capsid polyhedra.** Faces of the capsid polyhedra are shown with black edges and the colored edges give the dual graph. See the section on interaction cruciality.

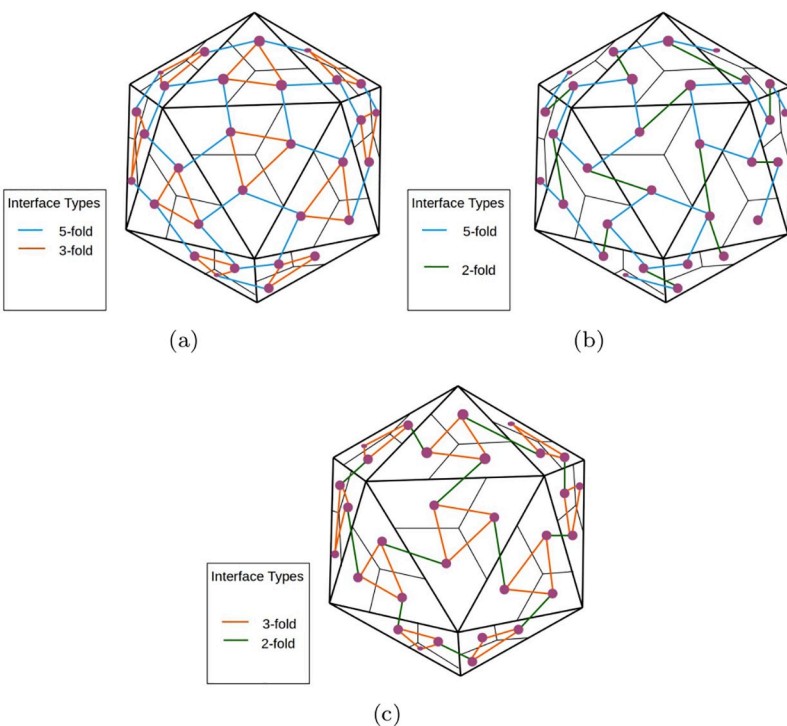

**Fig 7. *T* = 1 capsid polyhedra showing all 3 possible connectivity pathways.** See the section on interface cruciality.

Informally, a *connectivity pathway* corresponds directly to a minimal set of interfaces that a successfully assembled capsid must contain to even be a connected structure. It consists of the icosahedral symmetry classes of assembly trees that use only this minimal set of interfaces, weighted by the number of trees.

Given the dual graph $G = (V, E)$, of the capsid polyhedron (defined in the section on interaction cruciality), $E$ can be partitioned into sets $E_\iota$, one for each interface type $\iota$ (for $T = 1$ capsid polyhedra, there are 3 interface types and for $T = 3$ capsid polyhedra, there are 7 interface types as shown in Fig 6). A set $I$ of interface types is a *connectivity pathway* if the set of edges $E_I := \bigcup_{\iota \in I} E_\iota$, is a connected subgraph of $G$ and for each $\iota \in I$ $E_I \backslash E_\iota$ is not connected.

Given the small number of interface types for a capsid polyhedron of any $T$ number, we can find all connectivity pathways using a simple graph algorithm. Fig 7, shows 3 sets $I$ of interface types, for a $T = 1$ capsid polyhedron that correspond to connectivity pathways. Fig 8 shows 2 sets $I$ of interface types for a $T = 3$ capsid polyhedron, one of which is a connectivity pathway and one that is not. The *cruciality of an interface type* is the number of connectivity pathways containing that interface type.

## Two-scale prediction: Interaction cruciality at capsid-scale

We use two different types of two-scale predictions. The first prediction assumes that all interface types are equally important and is based only on the cruciality of interactions to interface types. For an interface of type $\iota$, the probability $P_\iota^r$ of breaking the interface when dropping an interaction $r$ is measured by the the cruciality bar-code: $(\mu_{minima}^r, \mu_{capsid}^r)$, as described in the section on interaction cruciality. Results validating these predictions are shown in the section on the output of cruciality prediction.

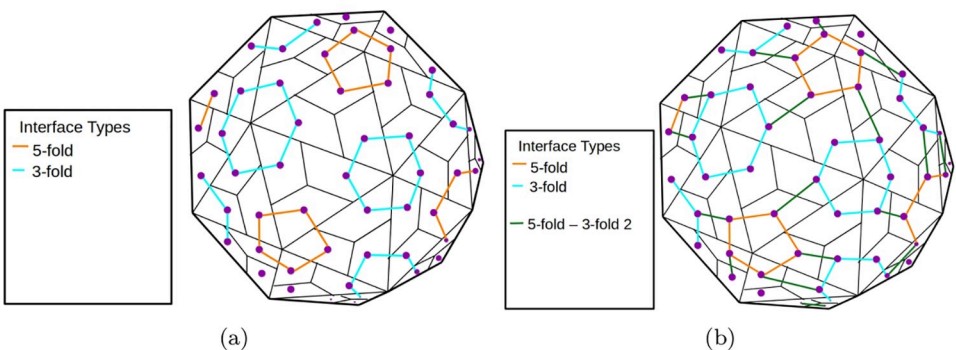

**Fig 8. $T = 3$ capsid polyhedra.** (a) Only the 5 fold and 3 fold interfaces are shown. This does not correspond to a connectivity pathway. (b) 5 fold, 3 fold and 5 fold-3 fold 2 interfaces are shown. This corresponds to a connectivity pathway. See the section on interface cruciality.

For the second two-scale prediction, we combine the interaction cruciality at the interface-scale and the interface cruciality at the capsid-scale using a statistical model as follows.

A simple linear model is used to learn the relative weights $a_\iota$, $b_\iota$, and $c_\iota$ for

$$P^r_\iota = \sigma(a_\iota \cdot \mu^r_{minima} + b_\iota \cdot \mu^r_{capsid} + c_\iota)$$

where $\sigma$ is the standard sigmoid or threshold function used in neural networks. The training data are obtained from site-directed mutagenesis results found in literature [14–20], that measure the effect of removing an interaction $r$ on capsid assembly.

In addition, we learn the relative weights of the two scales through a scalar parameter $w_\iota \in [0, 1]$ which represents the cruciality of an interface type $\iota$ to any connectivity pathway. The probability of breaking a connectivity pathway when an interaction is dropped is approximated by the equation:

$$C^r_p = 1 - \prod_{\iota \in p}(1 - w_\iota \cdot P^r_\iota) \tag{9}$$

For example, when $w_\iota = 0$, the corresponding term in Eq (9) vanishes, and breaking any interface of type $\iota$ has no effect on disrupting assembly. Conversely, when all $w_\iota$ are equal, the probability of disruption depends only on $P^r_\iota$'s, namely the cruciality of the interactions to interfaces and the number of connectivity paths in which an interface participates.

Putting these together, we get the *cruciality of an interaction r for capsid assembly* given by

$$H(r) = \sum_p C^r_p.$$

In this model, the parameters $a_\iota$, $b_\iota$, $c_\iota$, and $w_\iota$ are all unknown. We determine their value using simple machine learning. For a given partial order over the interactions $T = \{(r_i, r_j):r_i$ has bigger impact on the capsid assembly than $r_j\}$, the cruciality function $H$ should satisfy $H(r_i)>H(r_j)$. Towards this end, we design a loss function:

$$L = \sum_{(r_i,r_j)\in T} \sigma(H(r_i) - H(r_j))$$

where $\sigma$ is the standard sigmoid or threshold function used in neural networks. When the cruciality function $H$ satisfies the partial order, the loss function will be minimal. So the

parameters can be determined by evaluating

$$\arg\min_{a_l,b_l,c_l,w_l} L$$

Results validating these predictions are shown in the section on validating the second two-scale prediction.

## Results

The ab-initio computational predictions were blind to the results of the directed mutagenesis biophysical assays, noting that the biophysical assays for AAV2 were carried out by the Mavis Agbandje-Mckenna's lab [20] contemporaneously with development of our computational model and prediction. The section on experimental setup describes how the the in-vitro mutagenesis data were collected and processed from sources in literature. The section on input and output describes the input and output of the computational model used for prediction. The section on output of cruciality prediction describes the validation of the interaction cruciality prediction assuming that all interface types are equally important and is hence effectively based only on the interaction cruciality for interface assembly. The section on validating the second two-scale prediction describes the validation of the two-scale prediction which combines the interaction cruciality at the interface-scale and the interface cruciality at the capsid-scale using a statistical model.

### Experimental setup

We validate our prediction for AAV2 ($T = 1$), MVM ($T = 1$), and BMV ($T = 3$) viral capsids, using site-directed mutagenesis and biophysical assays to characterize the variants generated. AAV2, MVM, and BMV residues were selected based on their location in the 2-fold, 3-fold, and 5-fold symmetry-related interface of the viral capsid, and alanine scanning of all charged residues in the VP. The mutagenesis results used for validation were found in literature [14–20]. We note that the biophysical assays for AAV2 were carried out by the Mavis Agbandje-Mckenna's lab [20] contemporaneously with the development of our computational model and prediction.

The residues were classified by the yield of successfully assembled capsids compared to wild type after the mutation: a yield of 100% indicates that mutation has no effect on the assembly and the residue is marked *non-disrupt*; a yield of 0% indicates the assembly is completely disrupted, and the residue is marked as *disrupt* [14–19, 88–90].

All the in-vitro mutagenesis results used in this manuscript are detailed in Tables 1, 2 and 3, along with the sources of the data. Fig 9 shows these residues on a cartoon of their respective VP monomers.

### Input and output of computational prediction model

For the interface-scale prediction, we started from simplified potential energies designed from known X-ray structure of the VP monomers of each of the viruses and all their interfaces [91–93] We treated the participating VP monomers or dimers as single rigid motifs in the interface assembly systems.

The potential energy includes the hard-sphere potential between all atom pairs (one from each participating VP monomer or dimer) with the Van der Waals radius set to 1.2 Å. We used Lennard-Jones pair potentials, setting the energy difference of Eq (4) to $E_h - E_l = 0.997 kJ/Mol$ [94], and the weight in Eq (5) to $w(N_a) \approx 1.5^{N_a}$. An implicit solvent was assumed.

**Table 1. Mutagesis data used in this manuscript for AAV2 along with the sources of the data.**

| Residue | Mutagenesis result | Source |
|---------|--------------------|--------|
| 227 | Disrupt | [14] |
| 231 | Disrupt | [14] |
| 232 | Disrupt | [14] |
| 292 | Disrupt | [14] |
| 294 | Disrupt | [20] |
| 297 | Disrupt | [20] |
| 298 | Disrupt | [20] |
| 334 | Non-Disrupt | [16] |
| 337 | Non-Disrupt | [16] |
| 382 | Non-Disrupt | [18] |
| 389 | Non-Disrupt | [20] |
| 397 | Disrupt | [20] |
| 402 | Disrupt | [20] |
| 661 | Non-Disrupt | [20] |
| 692 | Disrupt | [20] |
| 694 | Disrupt | [20] |
| 696 | Disrupt | [14] |
| 704 | Non-Disrupt | [18] |
| 706 | Non-Disrupt | [18] |

Interface assembly landscapes were atlased using EASAL using the X-ray 3D structures of the participating VP monomers and dimers and the above-described pair potentials as input. Altogether 9 such atlases were obtained for different interface assembly systems for each of the $T = 1$ interface types and 31 such atlases for the $T = 3$ interface types, as described in the section on interaction cruciality. Each atlas computation takes no more than a couple of hours on a laptop with Intel Core i5-2500K @ 3.2 Ghz CPU and 8GB of RAM. Computations of cruciality required modification and analysis of each atlas for each interaction (approximately 20 per interface). Furthermore, we took into account the simultaneous disabling of all interactions involving a residue, as occurs in site-directed mutagenesis experiments. These analyses took microseconds.

**Table 2. Mutagesis data used in this manuscript for MVM along with the sources of the data.**

| Residue | Mutagenesis result | Source |
|---------|--------------------|--------|
| 55 | Disrupt | [17] |
| 129 | Disrupt | [17] |
| 153 | Disrupt | [17] |
| 168 | Non-Disrupt | [17] |
| 261 | Non-Disrupt | [20] |
| 302 | Disrupt | [19] |
| 507 | Non-Disrupt | [20] |
| 540 | Non-Disrupt | [20] |
| 543 | Disrupt | [17] |
| 546 | Disrupt | [17] |
| 567 | Disrupt | [19] |

**Table 3. Mutagesis data used in this manuscript for BMV along with the sources of the data.**

| Residue | Mutagenesis result | Source |
|---|---|---|
| 51 | Disrupt | [15] |
| 180 | Partial Disrupt | [15] |
| 181 | Disrupt | [15] |
| 182 | Disrupt | [15] |
| 183 | Partial Disrupt | [15] |
| 184 | Disrupt | [15] |
| 185 | Non-Disrupt | [15] |
| 188 | Partial Disrupt | [15] |
| 189 | Partial Disrupt | [15] |

In all cases, the interface-scale predictions were performed blindly without knowledge of in-vitro site directed mutagenesis results concerning assembly-driving interactions. In particular, for AAV2, mutagenesis results were only obtained subsequent to the interface-scale predictions. For MVM and BMV, the in-vitro site directed mutagenesis results were also gathered, subsequent to the interface-scale prediction, from multiple sources [14–19, 88–90]. For the training phase of the second two-scale prediction, less than half of the mutagenesis results were used, picking pairs of interactions marked disrupt and non-disrupt. Both training and learning phases took microseconds for each virus.

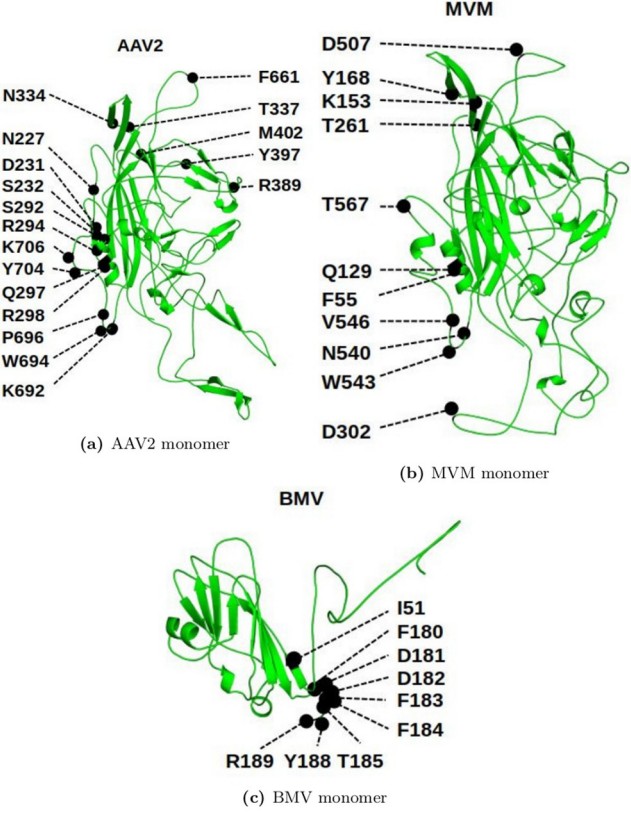

**Fig 9. AAV2 [91], MVM [92], and BMV [93] monomers showing the list of residues that were analyzed in this manuscript.**

**Justifying input assumptions.** There are two aspects of the Lennard-Jones potential that could possibly affect our prediction: (a) the energy difference $E_h - E_l$ and (b) the width we set for the discretized Lennard-Jones well. Our prediction method relies significantly more on the width of the discretized Lennard-Jones well rather than the actual energy difference $E_h - E_l$. The number of atom-pairs within the well essentially determines the quantity $N_a$ which is the deciding quantity in the computation of the normalized partition function affecting the cruciality prediction. However, since we know that the given capsid assembly configuration is feasible, the minimum pairwise distance (which in all cases turned out to be Van der Waals) forces the lower bound of the width of the discretized Lennard-Jones well. On the other hand, since the given capsid assembly configuration is a local energy minimum with (at least) 6 atom-pairs within the Lennard-Jones well, we can measure a natural upper bound value for the width of the discretized Lennard-Jones well (a larger value would permit a energy-neutral motion within the basin to a configuration with more atom-pairs in the Lennard-Jones well, and hence lower-energy, contradicting the minimality of the given capsid assembly configuration).

Although, theoretically, the potential energy should include the Lennard-Jones potential of all atom pairs, only the set of atom pairs that are close enough to interact and are conserved in related viruses have noticeable contribution to the configurational entropy. For the different types of interfaces (3 types for $T = 1$ and 7 types for $T = 3$), we determined such pairs of interacting residues (10-20 pairs for each interface), called the *candidate interactions* of each interface.

**Output of interface-based cruciality prediction.** As discussed in the section on interaction cruciality, for each interface assembly system $s$, we use the unweighted versions of the cruciality bar-code $(\mu_{minima}^{r,s}, \mu_{capsid}^{r,s})$ to predict the cruciality of an interaction to an interface. Fig 10(a) shows the plot of of the these two parameters for the interface assembly system $s$ being the 5-fold interface with VP monomers for BMV. Each row shows $\mu_{minima}^{r,s}$, $\mu_{capsid}^{r,s}$ and their ratio in two BMV 5-fold interface assembly systems (shown at the bottom right) where the interaction $r$ (which is the row label) is removed. The row labeled 'None' is the *wild type* assembly system where no interaction has been removed. The wild type system has been used to normalize the values of all the other rows. The rows are sorted according to the largest value of $\mu_{capsid}^r$.

Fig 10(b) plots the same parameters, but instead of considering VP monomers assembling at the 5-fold interface, we consider the assembly of a VP monomer and a VP dimer (as shown to the bottom right). As explained in the section on interaction cruciality, sterics play a larger role

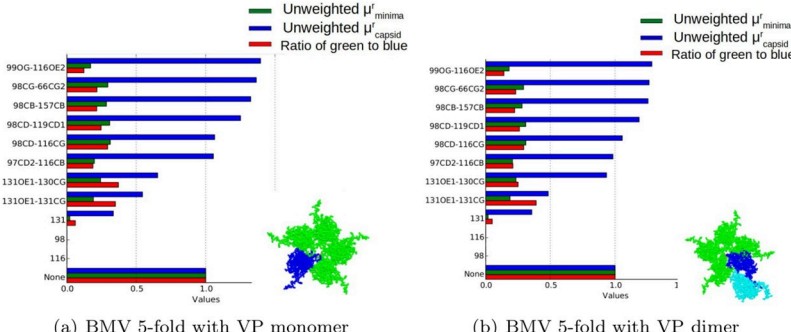

(a) BMV 5-fold with VP monomer          (b) BMV 5-fold with VP dimer

**Fig 10. Cruciality bar-codes.** Each row shows $\mu_{minima}^r$, $\mu_{capsid}^r$ and their ratio in two BMV 5-fold interface assembly systems (VP monomer-monomer and VP monomer-dimer shown at bottom right) where the interaction $r$—listed as the row label—is removed. The row labeled 'None' is the "wild-type" assembly system where no interaction has been removed, whose $v_{minima}$ and $v_{capsid}$ values have been normalized. The rows are sorted according to the largest value of $\mu_{capsid}$. See the section on output of cruciality prediction.

during the assembly of VP dimers than during the assembly of VP monomers. Note that certain interactions that had a lower value of $\mu_{capsid}$ in Fig 10(a) have a higher value in Fig 10(b). Since these plots merely illustrate our predictions without comparing them to mutagenesis data, the interested reader is referred to the following link https://urldefense.proofpoint.com/v2/url?u=https-3A__geoplexity.bitbucket.io_virusSuppInfo.html&d=DwIGaQ&c=sJ6xIWYx-zLMB3EPkvcnVg&r=kGsKDXNcJq1WRGevTkYaLhTe8S0Zrq5pLMzpMb45Vy0&m=RJ2zjPeU5q0XB4tHkgFcey-Z0oiqNxBsosEhffocKHs&s=GJyeBJcg4t2Sc8FaAu4t7P-RY_2U88_KxKF54dzWcPY&e=. for the complete set of such data, for all the interface assembly systems for all the viruses.

## Validating the first two-scale prediction

Fig 11 shows the cruciality bar-code of residues for those interface types (6 out of 13 interface types across the 3 viruses) for which there were sufficient in-vitro site directed mutagenesis results for validation and for which we were able to obtain cruciality predictions (see the discussion section). As explained in the section on interaction cruciality, the cruciality bar-code for an interface type is a cumulative value obtained from cruciality bar-codes computed for all the assembly systems at that interface.

Our interface-scale predictions were completely blind to the results of the directed mutagenesis biophysical assays, noting that the biophysical assays for AAV2 were carried out by the Mavis Agbandje-Mckenna's lab [20] contemporaneously with development of our computational model and prediction. Although generalizing an interface-scale prediction to the capsid level assumes the necessity of that interface for capsid assembly, our interface-scale predictions were validated successfully using mutagenesis data towards capsid assembly disruption. However, since this interface-scale prediction was part of a second prediction (see the section on interface cruciality) using statistical learning, that training data have been removed from Fig 11.

The points in Fig 11 represent candidate hotspot residues. The coordinate values at which they have been placed, are our computational cruciality predictions. The blue and red coloring of the points indicate residues found through mutagenesis to disrupt and not disrupt assembly respectively. The green circles are residues on which no mutagenesis was performed. The blue convex hull delineates the residues that are shown to disrupt, the red convex hull delineates the residues that are shown to not disrupt. Yellow delineates outliers. The sub-figures show that the predicted cruciality values of the residues that were later shown to disrupt assembly are linearly separated from the predicted cruciality values of the residues that do not disrupt assembly. I.e., the prediction convex hull formed by assembly-disrupting residues does not significantly intersect the convex hull formed by the non-assembly-disrupting residues. *Conversely, if the correlation between prediction and results were poor, there would not be such a linear separation (or separation of convex hulls).*

For a reader interested in independently running the EASAL software to reproduce our predictions, or in using other sources of experimental data to check our predictions, we refer to the link https://urldefense.proofpoint.com/v2/url?u=https-3A__geoplexity.bitbucket.io_virusSuppInfo.html&d=DwIGaQ&c=sJ6xIWYx-zLMB3EPkvcnVg&r=kGsKDXNcJq1WRGevTkYaLhTe8S0Zrq5pLMzpMb45Vy0&m=RJ2zjPeU5q0XB4tHkgFcey-Z0oiqNxBsosEhffocKHs&s=GJyeBJcg4t2Sc8FaAu4t7P-RY_2U88_KxKF54dzWcPY&e=, containing a complete set of such cruciality bar-code plots, individually for all the interface assembly systems, as well as the cumulative values for all the interface types, for the 3 viruses.

To compensate for the paucity of in-vitro site directed mutagenesis results, and to mitigate possible bias introduced when picking the candidate interactions, we added 2 more candidate

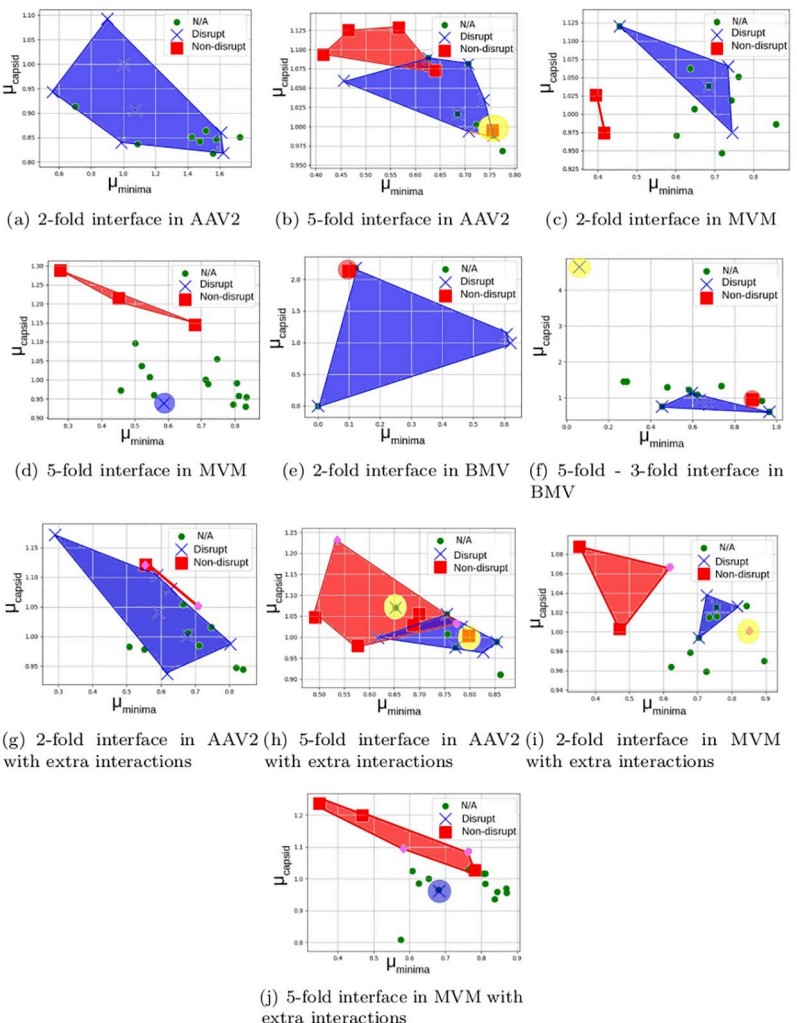

**Fig 11. Validation of direct interface-scale cruciality prediction: 2D plot of cruciality bar-codes for each interface.**
The blue cross marks and red squares are residues found, through mutagenesis, to disrupt and non-disrupt assembly.
The green circles are residues on which no mutagenesis was performed. The convex hulls are computational
predictions from our method. The blue convex hull delineates the residues that are shown to disrupt, the red convex
hull delineates the residues that are shown to not disrupt the assembly process, yellow convex hull delineates the
outliers. In (g)-(j), the pink diamonds are the extra interactions that were added to test for biases arising due to the
paucity of mutagenesis data. The black line shows a linear separation of the crucial and non-crucial residues. See the
section on validating the first two-scale prediction.

interactions to each interface. These interactions are unlikely to be crucial, since they were not
conserved across similar viruses. Atlases were regenerated for each interface assembly system,
with these additional interactions and the cruciality bar-codes were computed for all interac-
tions using the new atlases. The results for the two $T = 1$ viruses are shown as the last 4 figures
of Fig 11. Overall the added residues (red convex hull) fall outside the blue convex hull delin-
eating the residues shown to disrupt assembly.

## Validating the second two-scale prediction

Fig 12 shows, for AAV2, MVM and BMV, residues with their cruciality at the capsid-scale,
calculated using the statistical model described in the section on interface cruciality. The

**Fig 12. Validation of two-scale cruciality prediction using our statistical model for (a) AAV2, (b) MVM, and (c) BMV.** The residues listed higher in the table are computationally predicted through our method as more crucial and the ones lower in the table are predicted as less crucial. Experimental mutagenesis results are used to mark all the residues by color. Blue indicates that the residue disrupts assembly while red indicates that it does not. See the section on validating the second two-scale prediction.

correlation between our second two-scale prediction and in vitro site directed mutagenesis data is illustrated by the correlation between the prediction ranking of cruciality (top to bottom) and the mutagenesis data shown (using in-vitro mutagenesis and biophysical assays) extent of assembly disruption (color blue to red). More precisely, the residues listed on the top of the table are computationally predicted through our method to be more crucial, and the ones listed lower are predicted as less crucial. On the other hand, residues shown (using in-vitro mutagenesis and biophysical assays) to strongly disrupt assembly are blue and those that do not disrupt are red, with partial disruption indicated by the spectrum of colors in between. *Conversely, if the correlation between prediction and mutagenesis data were poor, the blues and reds would have been more interleaved.*

Fig 13 gives the full list of two-scale cruciality predictions using our statistical model for the three viruses. As before, the residues listed on the top of the table are computationally predicted through our method to be more crucial, and the ones listed lower are predicted as less crucial. The color codes have the same significance as explained in the previous paragraph. In addition, white indicates residues that do not yet have mutagenesis results for validation at the time of this writing.

As can be seen from Fig 13, the amount of available biophysical assays data for BMV was quite small when compared to the other two viruses. This paucity leads to the BMV results not showing the strong correlation seen in the results of the other two viruses. Despite this, it still correctly predict the cruciality for most of the residues for which in-vitro mutagenesis results are available.

## Discussion

Our prediction of crucial hotspot inter-atomic interactions between VP monomers for the assembly of icosahedral viral capsids in 3 viruses, starts from a candidate list of such interactions gleaned through sequence and structure conservation in evolutionarily similar viruses. This data was provided by Dr. Mavis Agbandje-Mckenna's lab. The crucial interaction prediction at the interface-scale is purely using statistical mechanics: it is not knowledge-based. We use an atlas (computed using the EASAL methodology) to approximate the changes in the partition function of the capsid. The prediction of interface cruciality at the capsid-scale uses an approximation of combinatorial entropy. One of our two types of predictions uses statistical learning to relatively weight the predictions at the two individual scales. site-directed mutagenesis and biophysical assays results validate both types of predictions.

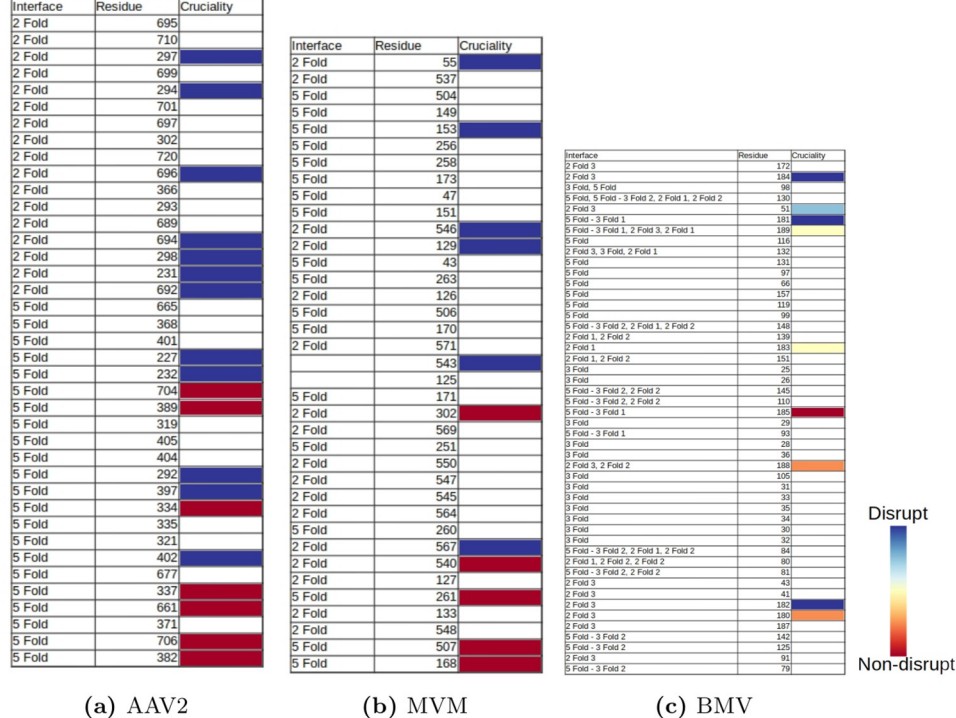

**(a)** AAV2          **(b)** MVM          **(c)** BMV

**Fig 13. Full list of two-scale cruciality predictions using our statistical model for (a) AAV2, (b) MVM, and (c) BMV.** The residues listed higher in the table are computationally predicted through our method as more crucial and the ones lower in the table are predicted as less crucial. Experimental mutagenesis results are used to mark all the residues by color. Blue indicates that the residue disrupts assembly while red indicates that it does not. White indicates residues that do not yet have mutagenesis results for validation at the time of this writing. See the section on validating the second two-scale prediction.

Kinetics have clearly been shown to affect many aspects of viral capsid assembly. While our method does not *separately* model kinetics, it combines interface-scale and capsid-scale analyses of free-energy and configurational entropy, which could affect kinetics. Our predictions do strongly rely on thermodynamics, specifically free energy. We believe that our results show that geometry and consequently configurational entropy may already be determinative in answering the narrow question of which inter-atomic interactions are more or less crucial to assembly, at least for the viral capsids studied here.

Besides a planned comparison of our interface-scale crucial interaction prediction method with a host of prevailing hotspot PPI prediction methods [26] on the SKEMPI database [43], there are several observations we made during the development of the method that may lead to future work.

*Additional interactions:* The candidate interactions that serve as the input to EASAL are hand picked and pre-screened. Some interactions are excluded because they are not likely to be crucial based some prior experience and some are excluded since no mutation on that residue is possible for now. This could potentially introduce bias in that the picked interaction are already likely more crucial than the others. In addition, since other non-crucial interactions also contribute to the potential energy, ignoring them will change the energy landscape. Using an extended set of candidate interactions as input would improve accuracy of prediction. Validation however would require more extensive mutagenesis results.

*Rigidity of the 3 and 5 fold interfaces:* As explained in the background section on configurational entropy, we decompose the viral capsids into interface *bi-assembly* systems involving two assembling units. However, for the 3 and 5 fold interfaces, simultaneous tri-assembly and pent-assembly should be considered. Better prediction could be obtained using newer variants of EASAL that handle more than two input assembling units [11].

*Omitted interfaces:* Our results in the section on output of interface-based cruciality prediction do not show the predictions of interaction cruciality for all interface types. Some of these omitted interfaces did not have mutagenesis results for validation, and have been included in the following link https://urldefense.proofpoint.com/v2/url?u=https-3A__geoplexity. bitbucket.io_virusSuppInfo.html&d=DwIGaQ&c=sJ6xIWYx-zLMB3EPkvcnVg&r= kGsKDXNcJq1WRGevTkYaLhTe8S0Zrq5pLMzpMb45Vy0&m=RJ2zjPeU5q0XB4tHkgFcey-Z0oiqNxBsosEhffocKHs&s=GJyeBJcg4t2Sc8FaAu4t7P-RY_2U88_KxKF54dzWcPY&e=. However, there are some interface that are not shown in the above link either, since we were unable to get any useful predictions for these interfaces. These include the 3-fold interfaces for $T$=1, $T$=3 virus and some 2-fold interfaces for $T$=3.

For 3-fold interfaces in AAV2, we could not obtain useful cruciality bar-codes or rankings due to the heavy influence of sterics caused by interdigitation. In addition, mutagenesis of the 3-fold interface interactions did not disrupt assembly. We do not believe that the removal of any of the 3-fold interactions causes assembly disruption. Most of the residues in the 3-fold interface of BMV cross-link to the RNA and hence have no effect on assembly. We conjecture that in these cases, the assembly proceeds primarily by 2-fold and 5-fold interface interactions. Trimer interdigitation contributes to post-assembly stability of the capsid.

*BMV results:* In the second two-scale prediction, the results for BMV do not exhibit the strong correlation seen in the results of AAV2 and MVM. While both AAV2 and MVM form empty capsid followed by the packing of the ssDNA genome, BMV instead co-assembles the capsid with the ssRNA genome, which plays an essential role in coordinating the assembly [95]. While empty BMV capsids can assemble in-vitro, the BMV wild-type has selected capsid proteins that co-assemble with the genome. Since our methodology only considers VP-monomers for crucial hotspot prediction, and doesn't take into account co-assembly, the BMV results may be skewed. However, we beleive that the bigger issue impacting the BMV results is the paucity of mutagenesis data available for the training model. Ranking of residues which currently do no have mutagenesis data for validation, are made available in the following link https://urldefense.proofpoint.com/v2/url?u=https-3A__geoplexity.bitbucket.io_virusSupp Info.html&d=DwIGaQ&c=sJ6xIWYx-zLMB3EPkvcnVg&r=kGsKDXNcJq1WRGevTkYaLh Te8S0Zrq5pLMzpMb45Vy0&m=RJ2zjPeU5q0XB4tHkgFcey-Z0oiqNxBsosEhffocKHs&s= GJyeBJcg4t2Sc8FaAu4t7P-RY_2U88_KxKF54dzWcPY&e=.

*Raw prediction data:* The raw prediction data for cruciality bar-codes, and ranking of residues which currently do no have mutagenesis data for validation, are available at https:// urldefense.proofpoint.com/v2/url?u=https-3A__geoplexity.bitbucket.io_virusSuppInfo. html&d=DwIGaQ&c=sJ6xIWYx-zLMB3EPkvcnVg&r=kGsKDXNcJq1WRGevTkYaLh Te8S0Zrq5pLMzpMb45Vy0&m=RJ2zjPeU5q0XB4tHkgFcey-Z0oiqNxBsosEhffocKHs&s= GJyeBJcg4t2Sc8FaAu4t7P-RY_2U88_KxKF54dzWcPY&e=.

The above-mentioned results, as well as all the results in the paper can be reproduced using the opensource EASAL software, curated by ACM TOMS in the collected algorithms of the ACM [10]. Software freely available on Bitbucket at https://urldefense.proofpoint.com/v2/url? u=http-3A__bitbucket.org_geoplexity_easal&d=DwIGaQ&c=sJ6xIWYx-zLMB3EPkvcn Vg&r=kGsKDXNcJq1WRGevTkYaLhTe8S0Zrq5pLMzpMb45Vy0&m=RJ2zjPeU5q0XB4tHk gFcey-Z0oiqNxBsosEhffocKHs&s=UvqE7o05ehbIXBe1Sgt920eHlxMg3vQCOuBWk0QU0l4 &e=. A user guide (https://bitbucket.org/geoplexity/easal/src/master/CompleteUserGuide.pdf)

and a video illustrating the features of the software https://cise.ufl.edu/%5C~sitharam/ EASALvideo.mpeg are also provided.

## Conclusion

In this paper we predict crucial inter-atomic interactions between VP monomers for the assembly of icosahedral viral capsids in 3 viruses, AAV2, MVM, and BMV. The crucial interaction prediction at the interface-scale uses an atlas generated with minimal sampling using the EASAL geometric methodology that relies on convexifying landscape regions using Cayley parameters. From the atlas, a cruciality bar-code approximates the changes in the partition function of the capsid assembly landscape when an interaction is removed. At the capsid-scale, an approximation of combinatorial entropy is used to predict the cruciality of interface types at the capsid scale. We use 2 two-scale methods to predict interface cruciality at the capsid scale. The first method is entirely blind to known site-directed mutagenesis and biophysical assay results, and assumes that each interface type is equally important for capsid assembly and only uses interaction cruciality at the interface scale to predict interaction cruciality at the capsid scale. The second method takes the variation among interface types into account, using statistical learning to relatively weight the predictions at the two scales. Site-directed mutagenesis towards assembly disruption are used to validate our predictions. The method, being computationally lightweight, rapid (100 to 1000 times faster than prevailing methods [11–13]), rigorous, and reliable, could be used to narrow down the field of candidate assembly-driving interactions for in-vitro experiments, or even computationally intensive in-silico experiments. For reproducibility, the reader can access and run the EASAL source code [12] with the help of descriptive papers [10, 11], user guide [96] and video tutorial [97], as well as all of our raw prediction data for cruciality bar-codes at URL https://urldefense.proofpoint.com/v2/url?u= https-3A__geoplexity.bitbucket.io_virusSuppInfo.html&d=DwIGaQ&c=sJ6xIWYx- zLMB3EPkvcnVg&r=kGsKDXNcJq1WRGevTkYaLhTe8S0Zrq5pLMzpMb45Vy0&m= RJ2zjPeU5q0XB4tHkgFcey-Z0oiqNxBsosEhffocKHs&s=GJyeBJcg4t2Sc8FaAu4t7P-RY_ 2U88_KxKF54dzWcPY&e=. This data includes EASAL predictions that could not be validated with the mutagenesis data we had access to, but could be checked against future mutagenesis experiments. At the interface-scale, the method is general enough to apply to any assembly system, in particular those that occur at various stages of the viral life-cycle, or during the action of tests and drugs. As Fig 2 shows, our single-scale methods can be mixed and matched piecemeal with prevailing methods to leverage complementary strengths (e.g., interface-scale hotspot predictions, or sequence and structure conservation, or extensive in-silico validation). We are unaware of any previous study that spans the range from multiscale statistical mechanical predictions of crucial hotspot inter-atomic interactions to validation using site-directed mutagenesis results, as opposed to Computational Alanine Scanning (CAS) and other computationally intensive Molecular Dynamics (MD) in-silico validations (explained in detail in the related work section).

## Acknowledgments

The authors thank Dr. Mavis Agbandje-Mckenna and Dr. Antonette Bennet for (1) compiling the data from the sources [14–20] in Tables 1, 2 and 3, (2) help with creating Fig 9, and (3) for providing us the candidate list of interactions, which served as the input to our model for crucial hotspot predictions (see the discussion section). We also thank them for contemporaneously carrying out the biophysical assays for site-directed mutagenesis on AAV2. Their results [20] along with others [14–19] helped validate our predictions.

We thank an insightful comment by an anonymous reviewer about our BMV predictions, now included in the discussion section.

## Author Contributions

**Conceptualization:** Meera Sitharam.

**Data curation:** Ruijin Wu.

**Formal analysis:** Meera Sitharam.

**Funding acquisition:** Meera Sitharam.

**Investigation:** Meera Sitharam.

**Methodology:** Ruijin Wu, Rahul Prabhu.

**Project administration:** Meera Sitharam.

**Software:** Ruijin Wu, Rahul Prabhu, Aysegul Ozkan.

**Writing – original draft:** Ruijin Wu, Rahul Prabhu, Meera Sitharam.

**Writing – review & editing:** Rahul Prabhu, Meera Sitharam.

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
