## [Decision Letter · Decision Letter 0]

1 Jun 2020

Dear Mr. Prabhu,

Thank you very much for submitting your manuscript "Rapid prediction of crucial hotspot interactions for icosahedral viral capsid self-assembly by energy landscape atlasing validated by mutagenesis" for consideration at PLOS Computational Biology.

As with all papers reviewed by the journal, your manuscript was reviewed by members of the editorial board and by several independent reviewers. In light of the reviews (below this email), we would like to invite the resubmission of a significantly-revised version that takes into account the reviewers' comments, in particular all the comments from reviewer 1.

We cannot make any decision about publication until we have seen the revised manuscript and your response to the reviewers' comments. Your revised manuscript is also likely to be sent to reviewers for further evaluation.

Sincerely,

David van der Spoel

Associate Editor

PLOS Computational Biology

Arne Elofsson

Deputy Editor

PLOS Computational Biology

Reviewer's Responses to Questions

**Comments to the Authors:**

Reviewer #1: In the manuscript “Rapid prediction of crucial hotspot interactions for icosahedral viral capsid self-assembly by energy landscape atlasing validated by mutagenesis”, Wu et al introduce a computational method to predict if mutations in capsid proteins can disrupt viral capsid assembly. There are currently no computational methods to assess this efficiently, and the research is significant. However, there are several major issues in the manuscript that challenged a rigorous scientific assessment of the research. The computational and experimental methods are incomplete, the comparison with experimental results is unclear, and the methodology introduces strong approximations that are not discussed with rigor. Below I listed my specific comments regarding these major issues as well as a number of minor issues.

Major issues:

1) The authors claim that the computational approach is validated experimentally (see, e.g., lines 119–121), but this comparison is unclear in the manuscript. I have a number of questions regarding this issue:

• Were the experiments carried in this research work, or were they published previously?

• Section 3.1 Experimental setup provides limited information about the experiments (lines 437–448). What where the mutations done for each virus and what were the assembly results of these experiments?

• Figures 10 and 11 provide the “validation” of the computational method and experiments, but it was not clear what was the prediction from the computational method, what was the experimental observations, and how they were compared.

2) Some parts of the computational method are described in great detail, but several elements that are key to judge the application of this method to the three viruses studied are missing:

• How were the 3D atomic models of the assembled viruses used to define the discretized Lennard-Jones potentials?

• How sensitive are the results of the computational method to variations to the Lennard-Jones pair potential energy difference?

• How sensitive are the results of the computational method to variations in the criteria of assignment of the Lennard-Jones discrete potential to atom pairs?

3) The methodology assumes a quasi-equilibrium pathway of assembly, where the partial capsids can be mapped onto the final capsid configuration. This is a strong approximation that disregard thermodynamic and kinetic mechanisms established previously

• The limitations of the model should be put in context with the current knowledge of assembly pathways (see for example, Reguera et al Soft Matter 2019, Ning et al Nature Communications 2016, Hagan and Elrad Bioph J 2010, and the references within).

Minor issues

1) Several terms like cruciality, subassembly, or atlasing are used early in the manuscript without having been properly defined. Some of the terms are defined later on in the manuscript, but it takes several pages. The authors should provide definitions for these terms early on in the manuscript.

2) The title of the manuscript claims that is a “rapid prediction” but this claim is based on a vague sentence referring to the computational time needed on a “laptop”. The specs of this laptop should be provided (CPU, RAM, number of processors…) so the community can properly judge this computational time.

3) The authors claim that “Icosahedral capsid self-assembly is poorly understood process” (lines 13 and 14). This is misleading. There are various principles (nucleation process, weak interactions, etc.) that are well established (see references given above). This includes the assumptions used in the computational method presented here. The authors should acknowledge this properly and clarify the aspects that remain open (like the multi-scale aspect).

4) Figure 1a is unclear. In a T=1 each monomer is involved equivalently in each icosahedral axis of symmetry. All proteins are equivalent in such capsid. Do the colors associated to each axis reflect domains in the monomers?

5) Review grammar throughout the text, for example (lines 13-14, lines 44-45, …)

Reviewer #2: The manuscript deals with computational modelling of the assembly of icosahedral viruses. Thereby, interface-scale and capsid-scale modelling approaches are combined. The approach uses the EASAL software of the same group. Overall, the is well written and provides detailed explanations of the complex methods.

The only point is that the validation of the prediction with AAV2, MVM and BMV mutagenesis results. The statement of strong correlation in section 3.3 is not really supported by Figure 11. The authors should elaborate this point more detailed or discuss it more cautiously.

**Have all data underlying the figures and results presented in the manuscript been provided?**

Reviewer #1: No: Lennard-Jones discrete force field for each virus studied; mutagenesis experiments

Reviewer #2: Yes

PLOS authors have the option to publish the peer review history of their article (what does this mean?). If published, this will include your full peer review and any attached files.

Reviewer #1: No

Reviewer #2: No
---

## [Decision Letter · Decision Letter 1]

10 Aug 2020

Dear Mr. Prabhu,

Thank you very much for submitting your manuscript "Rapid prediction of crucial hotspot interactions for icosahedral viral capsid self-assembly by energy landscape atlasing validated by mutagenesis" for consideration at PLOS Computational Biology. As with all papers reviewed by the journal, your manuscript was reviewed by members of the editorial board and by several independent reviewers. The reviewers appreciated the attention to an important topic. Based on the reviews, we are likely to accept this manuscript for publication, providing that you modify the manuscript according to the review recommendations.

Please address the comments by referee in the discussion section of the manuscript.

Sincerely,

David van der Spoel

Associate Editor

PLOS Computational Biology

Arne Elofsson

Deputy Editor

PLOS Computational Biology

[LINK]

Please address the comments by referee in the discussion section of the manuscript.

Reviewer's Responses to Questions

**Comments to the Authors: **

Reviewer #1: The authors have addressed all the major and minor concerns that I raised in my initial review. I reviewed the manuscript a second time, but I do not have any additional comments.

Regarding the weaker correlation for the BMV case, I would like the authors to consider the following observation: AAV2 and MVM both form and empty capsid followed by the packing of the ssDNA genome. BMV instead co-assembles the capsid with the ssRNA genome, which plays an essential role in coordinating the assembly (Comas-Garcia, Viruses, 11:253, 2019). Empty BMV capsids can assemble in vitro, but the BMV wild-type has selected capsid proteins to co-assemble with the genome. Since the algorithm relies exclusively on information about capsid proteins, the approach seems better suited for viruses that form procapsids rather than viruses that co-assemble with the genome. The information about potential hot spots taking into account co-assembly information is missing in the current approach and might explain the issues the BMV results.

**Have all data underlying the figures and results presented in the manuscript been provided?**

Reviewer #1: Yes

PLOS authors have the option to publish the peer review history of their article (what does this mean?). If published, this will include your full peer review and any attached files.

Reviewer #1: No
---

## [Editor Report · Decision Letter 2]

9 Sep 2020

Dear Dr. Sitharam,

Thank you very much for submitting your manuscript "Rapid prediction of crucial hotspot interactions for icosahedral viral capsid self-assembly by energy landscape atlasing validated by mutagenesis" for consideration at PLOS Computational Biology. As with all papers reviewed by the journal, your manuscript was reviewed by members of the editorial board and by several independent reviewers. The reviewers appreciated the attention to an important topic. Based on the reviews, we are likely to accept this manuscript for publication, providing that you modify the manuscript according to the review recommendations.

Sincerely,

David van der Spoel

Associate Editor

PLOS Computational Biology

Arne Elofsson

Deputy Editor

PLOS Computational Biology

[LINK]
---

## [Editor Report · Decision Letter 3]

22 Sep 2020

Dear Dr. Sitharam,

We are pleased to inform you that your manuscript 'Rapid prediction of crucial hotspot interactions for icosahedral viral capsid self-assembly by energy landscape atlasing validated by mutagenesis' has been provisionally accepted for publication in PLOS Computational Biology.

Best regards,

David van der Spoel

Associate Editor

PLOS Computational Biology

Arne Elofsson

Deputy Editor

PLOS Computational Biology

---

## [Editor Report · Acceptance letter]

13 Oct 2020

PCOMPBIOL-D-20-00706R3 

Rapid prediction of crucial hotspot interactions for icosahedral viral capsid self-assembly by energy landscape atlasing validated by mutagenesis

Dear Dr Sitharam,

I am pleased to inform you that your manuscript has been formally accepted for publication in PLOS Computational Biology. Your manuscript is now with our production department and you will be notified of the publication date in due course.

With kind regards,

Matt Lyles
